# Sarcopenic Dysphagia Revisited: A Cross-Sectional Study in Hospitalized Geriatric Patients

**DOI:** 10.3390/nu15122662

**Published:** 2023-06-07

**Authors:** Marcel Calles, Rainer Wirth, Bendix Labeit, Paul Muhle, Sonja Suntrup-Krueger, Rainer Dziewas, Gero Lueg, Ulrike Sonja Trampisch

**Affiliations:** 1Department of Geriatric Medicine, Marien Hospital Herne, Ruhr University Bochum, 44625 Herne, Germany; marcel.calles@elisabethgruppe.de (M.C.); rainer.wirth@elisabethgruppe.de (R.W.); gero.lueg@elisabethgruppe.de (G.L.); 2Department of Neurology, University Hospital Münster, 48149 Münster, Germanypaul.muhle@ukmuenster.de (P.M.); sonja.suntrup-krueger@ukmuenster.de (S.S.-K.); 3Department of Neurology and Neurorehabilitation, Klinikum Osnabrück, 49076 Osnabrück, Germany; rainer.dziewas@klinikum-os.de

**Keywords:** oropharyngeal dysphagia, muscle, neurogenic dysphagia, sarcopenia, swallowing disorder, older, geriatric

## Abstract

Oropharyngeal dysphagia (OD) is a frequent finding in older patients with potentially lethal complications such as aspiration pneumonia, malnutrition, and dehydration. Recent studies describe sarcopenia as a causative factor for OD, which is occasionally referred to as “sarcopenic dysphagia” in the absence of a neurogenic etiology. In most of the previous studies on sarcopenic dysphagia, the diagnosis was based only on clinical assessment. In this study, flexible endoscopic evaluation of swallowing (FEES) was used as an objective method to evaluate the presence of OD, its association with sarcopenia, and the presence of pure sarcopenic dysphagia. In this retrospective cross-sectional study, 109 acute care geriatric hospital patients with suspected OD received FEES examination and bioimpedance analysis (BIA) in clinical routine. 95% of patients had at least one neurological disease, 70% fulfilled the criteria for sarcopenia, and 45% displayed moderate or severe OD. Although the prevalence of sarcopenia and OD was high, there was no significant association between OD and sarcopenia. Considering these results, both the association between sarcopenia and OD and pure sarcopenic dysphagia appear questionable. Further prospective studies are needed to elucidate if sarcopenia is merely an epiphenomenon of severe disease or whether it plays a causative role in the development of OD.

## 1. Introduction

Oropharyngeal dysphagia (OD) is a frequent finding in older patients and is particularly associated with the occurrence of malnutrition, dehydration, and aspiration pneumonia. In addition, dysphagia is associated with difficulty in medication adherence [1] and increased health care costs [2].

Thus, early diagnosis and therapy play an important role in the prevention of these potentially lethal complications [3,4]. A large proportion of OD is etiologically due to neurological diseases such as stroke, Parkinson’s disease, or other neurodegenerative diseases [5]. Due to the high prevalence of OD in geriatric patients, its multifactorial genesis, and its severe consequences, OD has recently been declared a geriatric syndrome [6].

However, recent studies also describe the growing importance of sarcopenia as a cause or cofactor of OD [7,8,9,10,11,12,13,14]. Sarcopenia is a common disease in geriatric patients. Studies have shown that about 11–50% of people over the age of 80 are affected by sarcopenia [15,16]. Sarcopenia is defined as the simultaneous loss of muscle strength and muscle mass [17,18,19]. Corresponding cut-off values are defined in the European Consensus Definition (EWGSOP2) [20]. Sarcopenia is associated with an increased rate of falls, fractures, and other unfavorable outcomes, including mortality [21,22]. In particular, aspiration pneumonia in older adults is increasingly reported to be associated with sarcopenia-related OD [23].

To describe the relationship between sarcopenia and OD, the term “sarcopenic dysphagia” was first used in 2012, when a mild correlation between the results of a water-swallowing test and the mean upper arm circumference was found in older patients from Japan [24]. Meanwhile, other studies have confirmed this association, and the meta-analysis by Zhao et al. [7] indicates a significant association between sarcopenia and OD by combining the data of nine different studies (adjusted odds ratio: 4.06; 95% CI 2.27–7.29) [13,14]. The relationship between sarcopenia and dysphagia is explained in particular by the fact that generalized muscular atrophy also involves the swallowing muscles, leading to impaired motoric swallowing performance [9,10,25]. Age-related anatomical changes include a smaller cross-sectional area of masticatory muscles, increased lingual atrophy with fatty infiltration, and decreased muscle fiber diameter [26]. The volume of the geniohyoid muscle is significantly reduced in old compared to young subjects and in aspirators compared to non-aspirators in otherwise healthy older subjects [27]. Weak muscular tongue strength is the major contributor to impaired bolus propulsion and is significantly associated with aspiration status in older individuals. Moreover, hand grip strength as an indicator for systemic sarcopenia was significantly correlated with tongue strength [28].

The association of both findings, sarcopenia and OD, has now culminated in the definition of the new entity of sarcopenic dysphagia [7,8,9,10,11]. According to this definition, sarcopenic dysphagia is diagnosed if sarcopenia constitutes the sole cause of oropharyngeal dysphagia [10].

Overall, in most of the previous studies on sarcopenic dysphagia, the diagnosis of dysphagia was based only on clinical dysphagia assessment and not on videofluoroscopy or flexible endoscopic evaluation of swallowing (FEES) as objective methods. In addition, in most of the studies, potential underlying neurological conditions such as subcortical vascular encephalopathy (SVE) were not systematically excluded by cranial imaging, although SVE seems to play an important role in the etiology of OD [29]. Particularly, this may have led to a relevant underestimation of the neurological causes of OD. Therefore, the results must be replicated in studies with objective dysphagia assessment and more detailed neurologic assessment before adopting the concept of sarcopenic dysphagia.

According to Wakabayashi, sarcopenia must be the sole cause of dysphagia in order to diagnose sarcopenic dysphagia [10]. Based on our clinical experience, the association of OD with sarcopenia without an underlying neurological, otolaryngologic, or skeletal system disease is questionable. However, to confirm the hypothesis that OD is associated with sarcopenia and to explore the existence of pure sarcopenic dysphagia, we conducted the present study. In this retrospective study, we used FEES as an objective method to evaluate the presence of OD in clinically well-characterized hospitalized patients in a geriatric acute care ward. 

## 2. Materials and Methods

This retrospective observational cross-sectional study was performed in the acute care geriatric hospital department of Marien Hospital Herne, the university hospital of Ruhr University Bochum, Germany. Patients have been consecutively admitted to the geriatric acute care ward between January 2019 and February 2022. Inclusion criteria were age 65 years and older, complete data on body composition with bioelectric impedance measurement (BIA), hand grip strength, and FEES. The exclusion criterion was a palliative care situation. We did not obtain informed consent from study participants due to the retrospective design of the study.

The ethical committee of Ruhr University Bochum approved the study protocol (no. 21-7376-BR, accessed on 5 October 2021), and the study is registered at the German Clinical Trial Register (DRKS00029689, 8 August 2022).

The routine geriatric assessment comprised, i.e., the age of the patient (at the day of hospital admission): height in cm, measured weight in kg, activities of daily living (Barthel Index) [30], risk of sarcopenia (SARC-F) [31], risk of malnutrition according to the Mini Nutritional Assessment short-form (MNA-SF) [32], depressive symptoms according to the Depression in Old Age Scale (DIA-S) [33], frailty (FRAIL) [34], and cognitive status (Montreal Cognitive Assessment (MoCA) [35]. If the assessment with MoCA was not fully recorded for non-cognitive reasons, i.e., in the case of severe visual impairment, we extrapolated the result according to the number of questions answered. Two experienced physicians independently extracted data from routine geriatric assessments and neurological diseases retrospectively from medical records. A MoCA of 20–24 was classified as mild cognitive impairment (MCI), except other reasons for cognitive impairment, such as delirium, were recorded. If a MoCA below 20 points was not associated with delirium or dementia, this was categorized as a non-classifiable cognitive disfunction. Both MCI and non-classifiable cognitive disfunction were categorized as relevant neurological diseases in this context because they were proven to be associated with dysphagia [36,37]. In addition to neurological diseases typically associated with oropharyngeal dysphagia [38], delirium and SVE were also classified as relevant neurological diseases because they are highly associated with dysphagia [29,39]. 


**Assessment of sarcopenia**


In accordance with the revised European consensus on the definition and diagnosis of sarcopenia (EWGSOP2) [20], the measurement of body composition and hand grip strength is needed to define the presence of sarcopenia in patients. A medical professional measured body composition using bioimpedance analysis (BIA) with mBCA 525 (seca, Hamburg, Germany). The intra-assay reliability is R^2^ = 0.94 for extracellular water, R^2^ = 0.98 for fat-free mass, and R^2^ = 0.98 for total body water [40].

Measurements were performed in the supine position during the morning. Appendicular skeletal muscle mass (ASM) was calculated as the difference between skeletal muscle mass and the muscle mass value of the torso. The cut-off points for low muscle quantity in ASM/height2 are <7 kg/m^2^ for men < 5.5 kg/m^2^ for women [20]. 

Therapists assessed hand grip strength with a hand dynamometer (SAEHAN Handdynamometer Professional, SH5001), and it is defined as the maximum value in kg from three attempts with the dominant hand privileged. The examination took place in a sitting position on a chair. The patient assumes the following measuring positions: upright sitting position, feet placed flat on the floor, shoulder adducted and neutrally rotated, elbow with 90 degrees of flexion, and forearm in neutral position. The handle position of the dynamometer was 2 [41]. The cut-off points for low grip strength are <27 kg for men and <16 kg for women [20].

Patients were assigned to have sarcopenia if both low muscle quantity and low grip strength were present.


**Assessment of dysphagia**


FEES allows an objective assessment of the swallowing function [20,42,43] and is considered, along with videofluoroscopy, the gold standard [42,43,44]. If a patient presented with clinical signs of oropharyngeal dysphagia or dysphagia was suspected due to the underlying disease, FEES was performed. A speech-and-language therapist (SLT) and a FEES-certified expert physician carried out the FEES with an ENF-VH2-Laryngoscope by Olympus (Hamburg, Germany) and the video documentation system by Rheder/Partner GmbH (Hamburg, Germany) according to a predefined protocol. After anatomical and functional judgment of all structures, three trials of each—green jelly, green-dyed sirup like thickened water, green-dyed liquid water, and white bread—of approximately 3 × 3 × 0.5 cm were given. The following findings were assessed for each consistency: premature bolus spillage, pharyngeal residue, laryngeal penetration, and aspiration. The severity of dysphagia was classified according to an established 4-grade scale [45,46]: none of the above-mentioned findings = normal; premature bolus spillage or residue = mild; penetration or aspiration of one consistency = moderate; penetration or aspiration of more than one consistency = severe. According to the high prevalence of moderate residue in geriatric patients, we defined the result of FEES for the analysis as a binary variable, with normal swallowing and mild dysphagia as not clinically relevant (no) and moderate and severe dysphagia as clinically relevant dysphagia (yes).


**Statistical analysis**


We computed the statistical analysis using IBM SPSS Statistics (Version 28, Armonk, NY, USA) with means and standard deviations (SDs) for continuous data and n (%) for categorical variables. 

We performed a chi-square test to determine the presence of dysphagia (yes or no) and sarcopenia (yes or no). Further statistical data analysis consisted of multiple logistic regression with dysphagia (yes or no) as an outcome variable. Analyses were performed using dysphagia (yes) as a reference. Possible confounder variables were selected a priori as the most important known causal and conditional risk factors or variables significantly different between groups. They were classified as indicator variables (sex, sarcopenia, pneumonia during hospital stay, non-classifiable cognitive disfunction, Barthel-Index separated by median). Corresponding odds ratios (OR and their 95% CI) were calculated. We accepted statistical significance at the two-sided 0.05 level, and all CIs were computed at the 95% level.

## 3. Results

Due to missing and some implausible BIA measurements, the study population comprised 109 patients (n = 43, 39% women) with a mean age of 81.3 ± 6.5 years. The mean MoCA score was 16.2 ± 5.5, and the mean Barthel index was 42.1 ± 19.6 points. 97 patients (89%) were categorized as frail. 104 patients (95%) had at least one underlying neurological disease. Table 1a (continuous variables) and Table 1b (categorical variables) show the characteristics of the study population.

### 3.1. Sarcopenia and Dysphagia

106 patients (97%) had low muscle quantity according to ASM/height2 (mean ASM/height 4.0 ± 1.2). 78 patients (72%) show low hand grip strength (mean 16.5 ± 11.2 kg). Seventy-six patients (70%) fulfilled the criteria of sarcopenia according to the EWGSOP2 consensus definition [20].

According to the FEES results, 49 patients (45%) had moderate or severe dysphagia with clinically relevant airway compromise (Table 1a,b). 

Two patients (4.1%) of 49 with dysphagia did not show any underlying neurological disease. One of these patients suffered from Zenker’s diverticulum and one from severe chronic obstructive pulmonary disease (COPD) as potential explanations for dysphagia. Only the patient with COPD also suffered from sarcopenia.

About one third of all patients (n = 31, 28% of all study patients) had both dysphagia and sarcopenia. 44 patients with no or mild dysphagia (40%) had sarcopenia. The chi-square test showed no significant association between sarcopenia and dysphagia (OR 0.63, 95% CI 0.28–1.42, *p* > 0.05) (Table 2). 

### 3.2. Multiple Logistic Regression

The multiple logistic regression analyses with moderate and severe oropharyngeal dysphagia as dependent variables did not show significant associations in the crude analyses with sarcopenia and gender. The presence of dysphagia was not associated with sarcopenia, with odds ratios (OR) of 0.59 (95% CI 0.26–1.35) and female sex 0.48 (95% CI 0.22–1.08), respectively. 

After adjustment for the covariates, estimates were slightly attenuated (Table 3). We found a significant inverse association between dysphagia and non-classifiable cognitive dysfunctions (OR 0.30, 95% CI 0.10–0.87).

Sensitivity analyses with the replacement of sarcopenia by low muscle quantity and low grip strength separately showed no relevant differences. 

## 4. Discussion

Assuming that the development of sarcopenia may also involve swallowing muscles, many studies have previously demonstrated a link between sarcopenia and oropharyngeal dysphagia [7,8,9,10,11,12,13,14,24]. The meta-analysis by Zhao et al. calculated the risk of dysphagia for sarcopenic older persons with an OR of 4.06 [7]. Consequently, dysphagia was observed in sarcopenic patients without obvious neurological disease, which led to the term sarcopenic dysphagia [10,12,47]. In the present study, we could not demonstrate a significant association between sarcopenia diagnosed by BIA and hand grip strength and OD confirmed by FEES in a population of geriatric hospitalized patients with suspected dysphagia. Although both the prevalence of sarcopenia (70%) and OD (45% moderate and severe) was high, we could not show an association of OD with sarcopenia. Even if sarcopenia were replaced by its components, appendicular muscle mass and handgrip strength, no association was detectable. In addition, only two patients with moderate or severe dysphagia displayed no neurological diseases, with only one of them suffering from sarcopenia. 

Considering these results, both the association of sarcopenia and dysphagia and pure sarcopenic dysphagia in the absence of neurological disease appear questionable. In this context, it should be mentioned that most of the previous studies on OD and sarcopenia were performed on study populations in the rehabilitation sector. Cranial imaging was usually not available in these studies, so subcortical vascular encephalopathy (SVE) could not be detected as a possible cause of dysphagia, although the strong association between SVE and OD has been confirmed in many studies, as demonstrated by the meta-analysis by Alvar and colleagues [29]. The same is true for the association of dementia [4,48,49], mild cognitive impairment [36,37,50], and delirium [29,39,51] with OD, which has not been considered an underlying neurological disease associated with OD in most studies. In addition, it should be mentioned that objective dysphagia diagnosis using videofluoroscopy was implemented only in one of the previous studies considered [52]. All other studies used clinical assessment tools for diagnosing dysphagia, which display lower validity than endoscopic assessment [38]. 

Overall, the data presented here suggest a high proportion of sarcopenic patients in our study, which was significantly higher than in previous studies about the prevalence of sarcopenia in geriatric hospitalized patients [53,54,55,56]. Most likely, this is demonstrating the poor health condition of geriatric hospital patients with obvious or suspected dysphagia. However, measuring muscle strength at the beginning of an acute hospital treatment, which is done as a component of our standard assessment, is somehow problematic and a limitation of our study because the prevalence of low muscle strength in the sense of sarcopenia is probably overestimated. 

We think that further studies with a prospective methodology are needed to decide if sarcopenia is just an epiphenomenon of severe neurological disease or if it plays a causative role in the evolution of oropharyngeal dysphagia. In healthy older adults, it has already been demonstrated that parameters of muscle mass and function do not play a major role in the evolution of presbyphagia, the age-related change in swallowing [57]. In this study, rather than reflecting pharyngeal sensory function, parameters were associated with presbyphagia. However, it could be that this is different in diseased people. 

In addition, the reaction of swallowing muscles to aging and disease may differ from that of other skeletal muscles, which has not yet been studied in detail. A well-conducted study with FEES by Sporns et al. showed that premorbidly reduced swallowing muscle volume predisposes to more severe dysphagia after acute stroke [58]. However, in a multivariate regression analysis, only an association between age and dysphagia severity could be determined, not an independent association between atrophy of swallowing muscles and the degree of swallowing impairment. In combination with our study, the atrophy of swallowing muscles and of appendicular muscles appears to be a cofactor of aging and disease and not an independent risk factor for OD. Up until now, the relationship between systemic sarcopenia and the specific loss of swallowing muscle mass and function has not been thoroughly evaluated. In a histopathological study, Rhee et al. demonstrated that thinning and death of striated muscle fibers occur more frequently in the larynx and pharynx than in other parts of the body. Pharyngeal muscles are tonically active, have a high degree of elasticity, do not develop maximal tension, and are composed of a mixture of slow- and fast-twitch fibers, with the former predominating. A continuous or tonic action mode of a muscle may be a factor connected with atrophy in older subjects [59]. Thus, manifestations of sarcopenia may differ between various muscle groups, and sarcopenia in swallowing muscles certainly deserves further research. 

Two of our results may appear surprising. First, in our regression analysis, no significant association of OD with neurological disease was detectable. Due to the fact that a neurological disease with suspected dysphagia was the main clinical indication for the FEES examination, 95% of all patients suffered from a neurological disease with or without dysphagia. Therefore, differences concerning neurological disease cannot be detected in this cohort. Second, non-classifiable cognitive disfunction was significantly associated with the absence of dysphagia, which appears to be counterintuitive. However, cognitive disfunction after exclusion of dementia, mild cognitive impairment, and delirium indicates a low probability of a neurodegenerative disease potentially leading to dysphagia. It has previously been demonstrated that cognitive disfunction during an acute disease in geriatric hospitalized patients frequently changes during the course of the disease, independently from delirium [60].

Taken together, our results do not indicate that sarcopenia is not relevant for dysphagia patients. We are confident that all the negative consequences of sarcopenia on the risk of falls, fractures, and mortality [16,17,21,22] are equally relevant for dysphagia patients. In addition, weakness of muscles involved in coughing and breathing probably plays a major role in how dysphagia patients may cope with aspiration [23].

## 5. Conclusions

This study could not confirm a significant association between sarcopenia according to EWGSOP2 and oropharyngeal dysphagia according to FEES in geriatric hospitalized patients. In addition, the entity of sarcopenic dysphagia appears questionable because all participants had other medical conditions that explained dysphagia. Future research should focus on longitudinal changes in swallowing muscles to determine to what extent they are affected by age- and disease-related muscle decline and how this decline affects swallowing function.

## Figures and Tables

**Table 1 nutrients-15-02662-t001:** Characteristics of the study participants in total and according to dysphagia status.

(a)											
	Total n = 109			No or Mild Dysphagia n = 60 (55%)			Moderate or Severe Dysphagia n = 49 (45%)			*p*	
	Mean ± SD	Min	Max	Mean ± SD	Min	Max	Mean ± SD	Min	Max	*t*-test	
Age in years at the day of admission	81.3 ± 6.5	65	96	81.3 ± 6.1	69	96	81.3 ± 7.0	65	95	0.99	
Height (m)	1.68 ± 0.09	1.45	1.90	1.67 ± 0.09	1.45	1.90	1.70 ± 0.1	1.52	1.88	0.92	
Weight (kg)	70.1 ± 15.4	34.6	123.0	70.8 ± 17.5	36.0	123.0	69.1 ± 12.6	34.6	96.0	0.57	
Body mass index (kg/m^2^)	24.8 ± 5.1	14.4	41.1	25.4 ± 5.6	15.0	41.1	24.0 ± 4.3	14.4	37.5	0.15	
Fat-free mass (kg)	49.0 ± 10.9	28.6	75.3	48.4 ± 11.1	28.6	72.7	49.7 ± 10.8	28.8	75.3	0.53	
Fat mass (kg)	21.7 ± 12.5	0.7	62.5	23.2 ± 13.3	1.6	62.5	19.8 ± 11.4	0.7	51.2	0.83	
Skeletal muscle mass (kg)	19.0 ± 6.7	5.9	45.4	18.8 ± 6.4	7.2	32.6	19.1 ± 7.1	5.9	45.4	0.86	
Appendicular Skeletal Muscle Mass (kg)	11.4 ± 4.1	3.9	30.8	11.3 ± 3.8	4.4	22.9	11.5 ± 4.5	3.9	30.8	0.80	
Appendicular Skeletal Muscle Mass/height^2^	4.0 ± 1.2	1.4	9.3	4.0 ± 1.1	2.0	8.5	3.9 ± 1.3	1.4	9.3	0.84	
Hand grip strength (kg)	16.5 ± 11.2	0.1	50.0	15.2 ± 10.5	0.1	38.0	18.2 ± 11.8	0.1	50.0	0.17	
MoCA (n = 77)	16.2 ± 5.5	1	29	16.6 ± 4.4	6	27	15.7 ± 6.8	1	25	0.44	
Barthel-Index	42.1 ± 19.6	0	95	46.7 ± 19.6	0	95	36.4 ± 18.2	0	85	**0.01**	
**(b)**											
	**Total** **n = 109**			**No or Mild Dysphagia** **n = 60 (55%)**			**Moderate or Severe Dysphagia** **n = 49 (45%)**			**OR**	**95% CI**
	**number**	**%**		**number**	**%**		**number**	**%**			
Female	43	39		28	47		15	31		0.50	0.23–1.11
Sarcopenia	75	70		44	75		31	63		0.63	0.28–1.42
Dysphagia severity											
normal swallow	22	20		22	37						
mild	38	35		38	63						
moderate	33	30					33	67			
severe	16	15					16	33			
Low muscle quantity (ASM/height2)	106	97		59	98		47	96		0.40	0.04–4.53
Low hand grip strength (kg)	78	72		46	77		32	65		0.57	0.25–1.33
Frailty (n = 102)										1.3	0.21–8.12
pre-frail	5	5		3	5		2	4			
frail	97	89		52	87		45	92			
missing	7	6		5	8		2	4			
Risk of Sarcopenia (SARC-F questionnaire) (n = 101)										1.34	0.36–5.08
low risk	10	9		6	10		4	8			
high risk	91	84		48	80		43	88			
missing	8	7		6	10		2	4			
Mini nutritional assessment short form (MNA-SF) (n = 103)										1.04 *	0.47–2.31
normal nutritional status	2	2		1	2		1	2			
at risk of malnutrition	38	35		21	35		17	35			
malnourished	63	58		34	57		29	59			
missing	6	5		4	6		2	4			
Barthel-Index											
<40 pts	45	41		19	32		26	53		**2.44**	**1.12–5.33**
Depression in Old Age Scale (DIA-S) (n = 88)										0.63 **	0.27–1.47
no depressive symptoms	29	27		15	25		14	29			
mild to moderate depressive symptoms	15	14		8	13		7	14			
severe depressive symptoms	44	40		28	57		16	33			
missing	21	19		9	15		12	24			
Death during hospital stay	3	3		1	2		2	4		2.51	0.22–28.54
Pneumonia during hospital stay	23	21		8	13		15	31		**2.87**	**1.10–7.50**
Pneumonia during last year	7	6		3	5		4	8		1.69	0.36–7.93
At least one underlying neurological disease (multiple answers are possible)	104	95		57	95		47	96		1.24	0.20–7.71
dementia	50	46		24	40		26	53		1.70	0.79–3.64
stroke during hospital stay	28	26		18	30		10	20		0.60	0.25–1.45
stroke in history	30	28		20	33		10	20		0.51	0.21–1.23
Parkinson’s syndrome	17	16		9	15		8	16		1.11	0.39–3.12
critical illness disease	5	5		2	3		3	6		1.89	0.30–11.80
delirium	18	17		8	13		10	20		1.67	0.60–4.61
marked vascular encephalopathy	23	21		10	17		13	27		1.81	0.71–4.57
mild cognitive impairment	16	15		10	17		6	12		0.70	0.23–2.08
non-classifiable cognitive disfunction	25	23		19	32		6	12		**0.30**	**0.11–0.83**
myasthenia gravis	1	1		0	0		1	2		N/A	
intracranial hemorrhage	2	2		2	3		0	0		N/A	
amyotrophic lateral sclerosis	1	1		0	0		1	2		N/A	
traumatic brain injury	1	1		0	0		1	2		N/A	

Significant differences are highlighted in bold. n number of participants, *SD* standard deviation, *min* minimum, *max* maximum, *m* meter, *kg* kilogram, *MoCA* Montreal Cognitive Assessment, *kg* kilogram, *ASM* Appendicular Skeletal Muscle Mass. OR: odds ratio; CI: confidence interval. * MNA-SF as binary variable: normal nutritional status; at risk of malnutrition; and malnourished. ** DIA-S as binary variable: no, mild to moderate depressive symptoms, severe depressive symptoms.

**Table 2 nutrients-15-02662-t002:** Dysphagia and sarcopenia.

	Dysphagia		Total
Sarcopenia	No or Mild	Moderate or Severe	
no	16 (27%)	18 (37%)	34 (31%)
yes	44 (74%)	31(63%)	75 (69%)
total	60	49	109

**Table 3 nutrients-15-02662-t003:** Odds ratios (with 95% confidence intervals) for dysphagia.

Variable	OR	95% CI
Sarcopenia	0.49	0.20–1.21
female sex	0.63	0.26–1.54
Barthel Index < 40 pts.	2.31	0.95–5.58
Pneumonia during hospital stay	1.70	0.59–4.89
non-classifiable cognitive disfunction	**0.30**	**0.10–0.87**

Significant differences are highlighted in bold. Multiple logistic regression analysis with dysphagia as a dependent variable and all other variables as independent variables. OR: odds ratio; CI: confidence interval.

## Data Availability

Additional data are available from the corresponding author on reasonable request.

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
