# Peer review of "Sarcopenic Dysphagia Revisited: A Cross-Sectional Study in Hospitalized Geriatric Patients"

_nutrients, 2023, doi:10.3390/nu15122662_

Round 1

Reviewer 1 Report

Thank you for the opportunity to revise the current MS:

Please find my comments and suggestions below

Introduction:

Could you please define a study hypothesis ?

Methods:

Line 87: Typo, please amend accordingly.

Line 91: Why is this the only exclusion criteria ? Could you please explain this in detail.

Line 92: Please change to study participants

Line 166: Could you please add more details on BIA and handgrip assessments, like posture of participants, previous fluid intake, number of attempts etc.

Results:

Please report the intra-assay reliability of BIA assessments in you work.

Discussion:

Line 217 – 221: Could you please elaborate why ? Please see the following papers that might be of help

https://pubmed.ncbi.nlm.nih.gov/37208654/

https://pubmed.ncbi.nlm.nih.gov/28866357/

https://pubmed.ncbi.nlm.nih.gov/36708273/

https://pubmed.ncbi.nlm.nih.gov/34213693/

Line 244: State perhaps: Data presented here suggest ….

Conclusion:

This seems a bit vague, could you expand on you findings and support with some numbers ?

Also, could you give some recommendations for further research ?

Thank you for the opportunity to revise the current MS:

Please find my comments and suggestions below

Introduction:

Could you please define a study hypothesis ?

Methods:

Line 87: Typo, please amend accordingly.

Line 91: Why is this the only exclusion criteria ? Could you please explain this in detail.

Line 92: Please change to study participants

Line 166: Could you please add more details on BIA and handgrip assessments, like posture of participants, previous fluid intake, number of attempts etc.

Results:

Please report the intra-assay reliability of BIA assessments in you work.

Discussion:

Line 217 – 221: Could you please elaborate why ? Please see the following papers that might be of help

https://pubmed.ncbi.nlm.nih.gov/37208654/

https://pubmed.ncbi.nlm.nih.gov/28866357/

https://pubmed.ncbi.nlm.nih.gov/36708273/

https://pubmed.ncbi.nlm.nih.gov/34213693/

Line 244: State perhaps: Data presented here suggest ….

Conclusion:

This seems a bit vague, could you expand on you findings and support with some numbers ?

Also, could you give some recommendations for further research ?

Author Response

Dear reviewer,
thank you very much for your valuable reports. You will find my answer below each comment.
Review Report Form Reviewer #1
Thank you for the opportunity to revise the current MS:
Please find my comments and suggestions below
Introduction:
Could you please define a study hypothesis? 
Thank you for this remark. I’ve added it in line 80.
Methods:
Line 87: Typo, please amend accordingly.
I am not sure if I’ve found the correct typo. I’ve changed ; to ,
Line 91: Why is this the only exclusion criteria ? Could you please explain this in detail.
We excluded patients in a palliative care situation to prevent burdensome and medically unnecessary measurements in this group.
Line 92: Please change to study participants
See line 92
Line 166: Could you please add more details on BIA and handgrip assessments, like posture of participants, previous fluid intake, number of attempts etc. 
I’ve added detailed information on handgrip assessment in line 130 and BIA in line 124
Results:
Please report the intra-assay reliability of BIA assessments in you work.
I’ve added information on R2 in line 122.
Discussion:
Line 217 – 221: Could you please elaborate why ? Please see the following papers that might be of help 
See line 225 for clarification.
https://pubmed.ncbi.nlm.nih.gov/37208654/
https://pubmed.ncbi.nlm.nih.gov/28866357/
https://pubmed.ncbi.nlm.nih.gov/36708273/
https://pubmed.ncbi.nlm.nih.gov/34213693/
Line 244: State perhaps: Data presented here suggest …. 
Thank you, I’ve changed it, see line 253
Conclusion:
This seems a bit vague, could you expand on you findings and support with some numbers ?
To our opinion the conclusion should be a one or two sentence resume of the findings without numbers. The exact data can be red in the result section.
Also, could you give some recommendations for further research ?
Yes, see line 309.

Reviewer 2 Report

This is an interesting clinical and observational study, the data merit publication. The following points should be addressed in a revised version of the ms.

First, the study population is rather small, which limits general conclusions. Thus, the characteristics of the study population should be compared with a German reference population (e.g. see Nutrients 2020; 12:755).

Second, the definition of low skeletal muscle mass is based on data given in ref.20. However, estimates of skeletal muscle mass are method-specific. Thus, the cut offs used in the ms should be compared with the seca mBCA cut offs given in the reference cited above.

Third, the authors presented both, data on whole body SMM as well as on appendicular SMM. Since sarcopenia is characterized by a loss in whole body SMM this should be within the major focus of the ms. 

Fourth, unfortunately the authors did not provide data on FM and thus the SMM-FM ratio (which reflects the so-called capacity-load model) which is influenced by age and impacts disease prediction (see Public Health Nutrition 2015; 18: 1245-1254). These data should be added to the ms.

Fifth, the authors mix up qualitative data (e.g. data obtained by a nutritional risk score) with quantitative data. They should clearly differentiate between the risk of malnutrition and a quantitative estimate of malnutrition.

Author Response

Dear reviewer,
thank you very much for your valuable reports. You will find my answer below each comment.

Review Report Form #2
This is an interesting clinical and observational study, the data merit publication. The following points should be addressed in a revised version of the ms.
First, the study population is rather small, which limits general conclusions. Thus, the characteristics of the study population should be compared with a German reference population (e.g. see Nutrients 2020; 12:755).
We did not aim to investigate a representative population. We aimed to study a population with a high prevalence of both sarcopenia and dysphagia. That is why we performed this study in geriatric hospital patients.
Second, the definition of low skeletal muscle mass is based on data given in ref.20. However, estimates of skeletal muscle mass are method-specific. Thus, the cut offs used in the ms should be compared with the seca mBCA cut offs given in the reference cited above.
The consideration of method specific values is very valuable and we would be glad to be able to compare them with the seca mBCA cut offs given in the reference of Walowski et al. (2020). However, we are not sure if the proposed values can be applied to multimorbid and severely ill patients of very high age, even if they are available for the device used. In addition, the reference cut-offs for sarcopenia are not just percentiles but reference values reflecting a decline of mobility and other clinical consequences. 
Third, the authors presented both, data on whole body SMM as well as on appendicular SMM. Since sarcopenia is characterized by a loss in whole body SMM this should be within the major focus of the ms. 
The sarcopenia consensus definition offers both whole body and appendicular muscle mass as a measure of muscle mass. As appendicular mass is much more relevant for mobility and prognosis, we decided to use appendicular mass.
Fourth, unfortunately the authors did not provide data on FM and thus the SMM-FM ratio (which reflects the so-called capacity-load model) which is influenced by age and impacts disease prediction (see Public Health Nutrition 2015; 18: 1245-1254). These data should be added to the ms.
We have now added the data on FM to Table 1a. Since this paper is not primarily about nutritional medicine, we have deliberately omitted the presentation of the capacity-load model so as not to confuse readers. 
Fifth, the authors mix up qualitative data (e.g. data obtained by a nutritional risk score) with quantitative data. They should clearly differentiate between the risk of malnutrition and a quantitative estimate of malnutrition.
You are completely right to emphasize that MNA-SF is just a risk screening. However, we give data of MNA-SF just to characterize our study population. In all studies in a geriatric population it is useful to characterize the population with data of the geriatric assessment, because it enhances comparability of the population and reflex prognosis in this population much better than comorbidity. Nutritional state per se was not a topic of our study.

Reviewer 3 Report

Authors described a retrospective study of 109 subjects from one hospital concerning the potential causative sarcopenia factor for oropharyngeal dysphagia (OD). Also known as sarcopenic dysphagia.

The results indicated no association probably due to a difference (less objective) in methods to diagnose dysphagia described in previous literature.

It is an important observation for diagnostic purposes and future prediction or prevention/treatment strategies.

The manuscript is well written. No comments

Author Response

Dear reviewer,
thank you very much for your valuable reports. You will find my answer below each comment.

Review Report Form #3 
Authors described a retrospective study of 109 subjects from one hospital concerning the potential causative sarcopenia factor for oropharyngeal dysphagia (OD). Also known as sarcopenic dysphagia.
The results indicated no association probably due to a difference (less objective) in methods to diagnose dysphagia described in previous literature.
It is an important observation for diagnostic purposes and future prediction or prevention/treatment strategies.
The manuscript is well written. No comments

Dear Reviewer, thank you very much for this positive feedback. Your comments were very motivating.
